# Missing science: A scoping study of COVID-19 epidemiological data in the United States

**Rajiv Bhatia**[1]*, **Isabella Sledge**[2], **Stefan Baral**[3]

**1** Primary Care and Population Health, Stanford University, Stanford, CA, United States of America, **2** Ikigai Research, Los Angeles, CA, United States of America, **3** Department of Epidemiology, Johns Hopkins School of Public Health, Baltimore, MD, United States of America

* drajiv@stanford.edu

**Data Availability Statement:** All reviewed reports are publicly accessible via PubMed. An online supplement includes all data abstracted from reports meeting inclusion criteria.

## Abstract

Systematic approaches to epidemiologic data collection are critical for informing pandemic responses, providing information for the targeting and timing of mitigations, for judging the efficacy and efficiency of alternative response strategies, and for conducting real-world impact assessments. Here, we report on a scoping study to assess the completeness of epidemiological data available for COVID-19 pandemic management in the United States, enumerating authoritative US government estimates of parameters of infectious transmission, infection severity, and disease burden and characterizing the extent and scope of US public health affiliated epidemiological investigations published through November 2021. While we found authoritative estimates for most expected transmission and disease severity parameters, some were lacking, and others had significant uncertainties. Moreover, most transmission parameters were not validated domestically or re-assessed over the course of the pandemic. Publicly available disease surveillance measures did grow appreciably in scope and resolution over time; however, their resolution with regards to specific populations and exposure settings remained limited. We identified 283 published epidemiological reports authored by investigators affiliated with U.S. governmental public health entities. Most reported on descriptive studies. Published analytic studies did not appear to fully respond to knowledge gaps or to provide systematic evidence to support, evaluate or tailor community mitigation strategies. The existence of epidemiological data gaps 18 months after the declaration of the COVID-19 pandemic underscores the need for more timely standardization of data collection practices and for anticipatory research priorities and protocols for emerging infectious disease epidemics.

## Introduction

Efficient and effective pandemic control measures demand complete epidemiological data, including timely and precise parameters of infectious transmission, disease severity and disease burden. Insufficient or poor-quality data on transmission mechanisms, setting and activity specific risks, and intervention benefits can reduce the effectiveness, efficiency, and equity of a public health response [1].

**Funding:** The author(s) received no specific funding for this work.

**Competing interests:** The authors have declared that no competing interests exist.

Prior to COVID-19, the US Center for Disease Control and Prevention's (USCDC) pandemic strategy outlined specific data requirements for managing epidemics caused by novel respiratory viruses [2, 3]. The influenza H1N1 pandemic provided an opportunity to apply and evaluate these essential data elements [4–7], and many calls for their timely production accompanied the onset of the COVID-19 pandemic [8–10] (Table 1).

Key parameters of epidemic transmission include the incubation period (how long after infection symptoms appear), the clinical fraction, the generation interval (the time between a person becoming infected and subsequently infecting someone else), the infectivity period (when and how long an infected person can spread the illness), and the secondary attack rate (the risk of infection due to an infectious contact). Collectively, these parameters help predict the extent and intensity of epidemic transmission and help determine the feasibility and value of strategies for isolation and contact tracing [11, 12] as well as those for targeting groups or settings [13].

Measures of infection severity, such as infection-hospitalization and infection-fatality ratios, inform the social impact of epidemic transmission and help calibrate the scope and scale of control measures. Early estimates of infection severity come from surveillance systems while more reliable ones require prospective cohort studies [7].

Robust active and passive surveillance systems provide real-time monitoring of disease burden, including the incidence and prevalence of illness, numbers of people hospitalized, and infection-related deaths. Optimally, such surveillance data is disaggregated by population subgroup, setting, severity and patient characteristics to inform timely, targeted community mitigations as well as to allocate healthcare resources.

A scoping study aims to examine the extent, range, nature of evidence or research activities in a particular field and is useful for identifying data gaps [14]. Here, we report on a scoping study of COVID-19 epidemiologic data and epidemiologic research relevant for pandemic management in the United States. Our aims were to assess the completeness of this data during the first two years of the COVID-19 pandemic along with the responsiveness of governmental public health research. We first document authoritative estimates of key transmission and disease severity parameters in the U.S. and characterize publicly available disease surveillance data. We then identify and characterize epidemiologic investigations informing these parameters conducted and reported through November 2021 by U.S. governmental public health entities. Based on our review, we identify gaps in knowledge and missed research opportunities that may have weakened the U.S. pandemic response.

## Methods

We first examined the US CDC's public website for U.S. government estimates of COVID-19 transmission and infection severity parameters as well as surveillance indicators of infection and disease burden. We used the Internet Archive Wayback Machine (https://archive.org/web/), to assess the evolution of these parameters at different time-points during the pandemic. We characterized the scope of COVID-19 surveillance indicators of infection and disease burden with regards to measures and their person, time, and place aggregation on US CDC webpages at two timepoints—November 2020 and November 2021.

We next conducted a scoping review of observational epidemiology studies with outcomes related to COVID-19 infection transmission, severity, or disease burden which were conducted in U.S. settings, reported by authors with U.S. governmental public health affiliations, and published before November 30, 2021. The protocol for this scoping review adapted the Preferred Reporting Items for Systematic Reviews and Meta-analysis Protocols (PRISMAP) (S1 Protocol). We did not register this protocol prospectively. The protocol provides details of

**Table 1. Epidemiological data required for managing emerging respiratory virus epidemics.**

| Measure | Definition | Information value | Typical source |
|---|---|---|---|
| **Transmissibility** | | | |
| Basic reproductive number | The expected number of secondary cases directly generated by one case | Potential speed of epidemic growth | Calculated from contact rate, secondary infection risk, and infectious period or from the growth rate of the early disease incidence curve |
| Growth rate | Change per unit time (acceleration or deceleration) of the incidence rate | Current trajectory of epidemic growth | Population disease monitoring |
| Susceptible population | Proportion of the population who have immunity to the infection or to disease due to natural or acquired immunity | Targeting control measures | Studies of immunity such as seroprevalence of antibodies |
| Incubation period | Interval between infection and the development of symptoms | Timeframe for prevention of secondary infection | Transmission studies |
| Duration of infectiousness | Viral load and duration in symptomatic and asymptomatic people | Timeframe for prevention of secondary infection | Transmission studies |
| Serial interval | Interval between development of symptoms in a case and an infected contact | Necessary for capturing the $R_0$ | Transmission studies |
| Pre-symptomatic transmission | Proportion of infections spread by persons who appear well but are infected and later develop symptoms | Timeframe for prevention of secondary infection | Transmission studies |
| Secondary infection risk (SIR or alternatively, secondary attack rate) | Proportion of exposed people who become ill in a setting (household, school workplace) | Targeting control measures | Transmission studies |
| **Infection Severity** | | | |
| Symptomatic fraction | Proportion of infected people who become symptomatically ill | Estimating disease burdens and adopting a proportional response | Household or contact tracing transmission studies |
| Case hospitalization and fatality ratios | Ratio of identified cases to hospitalized and fatal cases | Estimating disease burdens and adopting a proportional response | Population surveillance |
| | | | Cohort studies |
| | | | Large transmission studies |
| Infection hospitalization and fatality ratios | Ratio of estimated infections to hospitalized and fatal infections | Estimating disease burdens and adopting a proportional response | Reported hospital data. Death records |
| Severity risk factors | Demographic, clinical, occupational, social, and environmental risk factors affecting vulnerability to severe disease outcomes | Targeting prevention measures | Case control studies |
| | | | Syndromic surveillance |
| **Disease Burden** | | | |
| Incidence rate | Number of new cases of illness, hospitalization or death in a population per unit time | Is the disease accelerating or slowing down and where | Syndromic surveillance |
| | | | Serial prevalence studies |
| | | | Disease hospitalization rates |
| | | | Disease mortality rates |
| Point or period prevalence | Proportion of the population that is a current case a point or period in time | Current level of active infection and transmission | Symptom and test-based surveys |
| Community attack rate / cumulative incidence | Number of new cases of disease during specified time interval | Population disease burden & remaining population susceptible | Cohort studies, statistical estimation |

our study eligibility criteria, search sources, search strategy, and data collection and management procedures.

Briefly, we utilized PubMed to identify potentially eligible studies with our desired target outcomes. We screened each PubMed search result in duplicate, including the title, abstract, and all author affiliations, to identify potentially eligible studies then read the full text of candidate studies to assess eligibility. We did not include studies reporting outcomes related to clinical management or pharmaceutical and vaccine interventions, nor did we include laboratory

studies of viral biology, phylogenetic studies, or synthetic modeling exercises. For included studies, we abstracted published information on authors' governmental affiliations, study methods, data source, data period, study setting, study population, and analytic outcomes.

We classified studies as descriptive or analytic based on their methods. We subcategorized descriptive studies as either case series or cluster investigations or as incidence studies. We subcategorized analytic studies as either cross-sectional, case-control, ecologic prospective cohort, or retrospective cohort designs. We categorized the study's primary data source as: passive or active surveillance program, administrative program records, medical or vital statistic records, serosurveys, questionnaire surveys, or original field data. We noted the end of each study's data collection period and classified a study's specific setting or subpopulation.

For descriptive incidence studies and all analytical studies, we assessed and noted whether the study reported any of the following outcomes: reproductive number or growth rate, secondary attack rate, incubation period, serial interval or generation time, symptomatic fraction, infection or case hospitalization ratio, infection, case, or hospital fatality ratio, incidence of infection, seroprevalence, case status, emergency department care, hospitalization or death, predictors of infection incidence, predictors of disease severity.

## Results

We found authoritative estimates for infection transmission and severity parameters published on the US CDC's Pandemic Planning Scenarios webpages. This webpage was published first in May 2020 and subsequently updated three times (Table 2) [15–18]. Point estimates for most parameters were consistent over the period although for some parameters, such as the clinical fraction, the confidence range remained wide. Most transmission parameter estimates were based on epidemiological studies conducted outside the U.S. using data collected in the first months of the pandemic. We did not find an authoritative estimate of the secondary attack rate for any setting. We also did not find applications of established US CDC's pandemic risk assessment tools, such as the Pandemic Severity Assessment Framework (PSAF).

The US CDC estimated case fatality and case hospitalization ratios in May 2020 but subsequently characterized disease severity measures as ratios of infection-fatality and hospital-fatality disaggregated by age (Table 2). The infection fatality estimates relied on European data while hospital fatality estimates relied on data from the US CDC COVID-NET active surveillance program.

The US CDC began weekly reporting of national-level COVID-19 surveillance measures in April 2020 on a page titled 'COVID View' [19]. In 2020, surveillance measures reported on the COVID View page included emergency department visits for coronavirus-like illness (CLI), COVID-19 test positive hospital admissions, and deaths from pneumonia, influenza, and COVID-19 [20] (Table 3). Data on hospital admissions came from a US CDC led active hospital-based surveillance implemented in 13 sub-state regions [21]. Mortality data came from the US National Vital Statistics System [22]. From August 2020, data on COVID-19 case and death incidence, disaggregated by age, sex, and race were also reported separately on the COVID Data Tracker page [23].

As illustrated in Table 3, over time, the scope and granularity of US CDC publicly reported surveillance data increased, and all data was accessible via the COVD Data Tracker page [24, 25]. National seroprevalence estimates from commercial laboratory sampling appeared after August 2020 and uniform national hospital admissions data appeared in December 2020. In 2021, the COVID Data Tracker page added data on vaccination effectiveness and included case and fatality rates for nursing home and health care personnel. On additional websites,

**Table 2. USCDC estimates for COVID-19 selected infection transmission and severity parameters.**

| | May 2020 | July 2020 | Sept 2020 | Mar 2021 |
|---|---|---|---|---|
| **Transmission Measures** | | | | |
| Reproductive number | 2.0 (2.0–3.0) | 2.5 (2.0–4.0) | 2.5 (2.0–4.0) | 2.5 (2.0–4.0) |
| Susceptibility | NA | NA | NA | NA |
| Mean incubation period | 6 days | 6 days | 6 days | 6 days |
| Mean serial interval | 6 days | 6 days | 6 days | 6 days |
| Percentage of transmission occurring prior to symptom onset | 40% | 50% (35–70) | 50% (30–70) | 50% (30–70) |
| Percent of infections that are asymptomatic | 35% (20–50) | 40% (10–70) | 40% (10–70) | 30% (15–70) |
| Relative infectiousness of asymptomatic individuals | 100% (50–100) | 75% (20–100) | 75% (25–100) | 75% (25–100) |
| Secondary Attack Rate | Not Estimated | Not Estimated | Not Estimated | Not Estimated |
| **Infection Severity Measures** | | | | |
| Symptomatic Case Hospitalization Ratio | 0–49 y 0.017 | | | |
| | 50–64 y 0.045 | | | |
| | 65+ y: 0.074 | | | |
| Symptomatic Case Fatality Ratio | 0–49 y 0.0005 | | | |
| | 50–64 y 0.002 | | | |
| | 65+ y: 0.013 | | | |
| Infection fatality ratio (%) | | 0.065 | 0–19 y: 0.003 | 0–17 y: 0.002 |
| | | | 20–49 y: 0.02 | 18–49 y: 0.05 |
| | | | 50–69 y: 0.5 | 50–64 y: 0.6 |
| | | | 70+ y: 5.4 | 65+ y: 0.09 |
| Hospital fatality ratio (%) | | 18–49 years: 2 | 18–49 years: 2.4 | 0–17 y: 0.7 |
| | | 50–64 years: 9.8 | 50–64 years: 10 | 18–49 y: 2.1 |
| | | ≥65 years: 28 | ≥65 years: 26.6 | 50–64 y: 7.9 |
| | | | | 65+ y: 18.8 |

USCDC published estimates of cumulative burdens of infection, symptomatic infection, hospitalization, and death [26] and estimates of pandemic period excess deaths [27].

Our scoping study report identification strategy returned 4823 unique publications of which, after screening and review, 283 met our inclusion criteria (Fig 1). We provide a table of the data abstracted from the included studies in supporting information (S1 Table).

The largest share of reports (61%) was published in the US CDC publication MMWR (Table 4). Most reports (74%) had authors affiliated with a combination of US CDC and State or local public health entities although a substantial number (25%) had authors affiliated only with the US CDC. The majority (57%) of reports were published in 2020. Seventy percent of studies utilized data collected before October 2020 with the remainder utilizing data collected between October 2020 and November 2021.

We categorized 180 of the 283 reports as descriptive studies [28–207]. One hundred and twenty-eight of the descriptive studies enumerated a series or cluster of confirmed COVID-19 cases in the general population within a particular setting or sub-population typically characterizing attack rates and the frequency of characteristics, exposures, and clinical outcomes among individual cases. Long-term care facilities, prisons, and social gatherings were the most common settings for descriptive reports. Congregate facility residents and staff and children and adolescents were the most frequently reported sub-populations. We characterized 51 of the 180 descriptive studies as incidence studies; these provided period, population, and geographically specific estimates of case, disease, seroprevalence, ED visits, or mortality incidence.

We categorized 103 of the 283 reports as being analytic [208–310]. Analytic studies most applied cross-sectional (41/103), ecologic (25/103) or retrospective (23/103) designs. The most

**Table 3. Scope of US CDC publicly reported Covid-19 disease surveillance data.**

| Surveillance Measure | Nov 2020 | | | Nov 2021 | | |
|---|---|---|---|---|---|---|
| | National | State | County | National | State | County |
| Surveillance Case incidence | X | X | X | X | X | X |
| Sex | X | | | X | X | X |
| Age group | X | | | X | X | X |
| Race/ ethnicity | X | | | X | X | X |
| Nursing home resident / staff | | | | X | X | |
| Health care personnel | | | | X | X | |
| Fraction of positive COVID-19 laboratory tests | X | X | | X | X | X |
| Share of emergency department visits for Influenza-like-illness | X | | | | | |
| Share of emergency department visits for Covid-like-illness | X | | | | | |
| Share of emergency department visits for confirmed COVID-19 | | | | X | X | |
| COVID-19 test positive new hospital admissions rate | X | | | X | X | X |
| Age group | X | | | X | X | |
| Mortality due to COVID-19 | X | X | | X | X | X |
| Sex | X | | | X | X | X |
| Age group | X | | | X | X | X |
| Race/ ethnicity | X | | | X | X | X |
| Nursing home resident / staff | | | | X | X | |
| Health care personnel | | | | X | X | |
| Infection-induced antibody seroprevalence | X | | | X | X | |
| Sex | | | | X | | |
| Age group | | | | X | | |
| Combined infection and vaccination-induced antibody seroprevalence | | | | X | X | |
| Sex | | | | X | | |
| Age group | | | | X | | |
| Race/ ethnicity | | | | X | | |
| Estimated cumulative Incidence of infections, hospitalizations, and deaths | | | | | | |
| Infections by age | | | | X | | |
| Symptomatic Infections by age | | | | X | | |
| Hospitalizations by age | | | | X | | |
| Deaths by age | | | | X | | |

common data sources for analytic studies were passive surveillance systems (35/103), field collected data (22/103), or seroprevalence surveys (20/103). Most (66/103) of the analytic studies examined general population subjects in community settings (74/103). Less commonly, analytic studies focused children and adolescents (10/103), healthcare workers or first responders (8/103), workers in other occupations (5/103), and residents and staff of long-term care facilities (5/103).

Nine analytic studies estimated the secondary attack rate [208, 223, 237, 235, 244, 248, 288, 297, 307]. Only one only of these estimated the SAR for non-household contacts [307]. Eight of the nine estimating a SAR utilized data from the first 6 months of the pandemic.

Two studies estimated the serial interval [273, 291]. Both utilized state-level data from the first six months of the pandemic. Two studies estimated the reproductive number utilizing early pandemic period data from one U.S. State [225, 291].

Twenty-five analytic studies provided estimates of period, population, and place incidence measures. Twenty of these studies provided period estimates of antibody seroprevalence. Three analytic studies estimated excess mortality at various timepoints [231, 269, 287]. One of

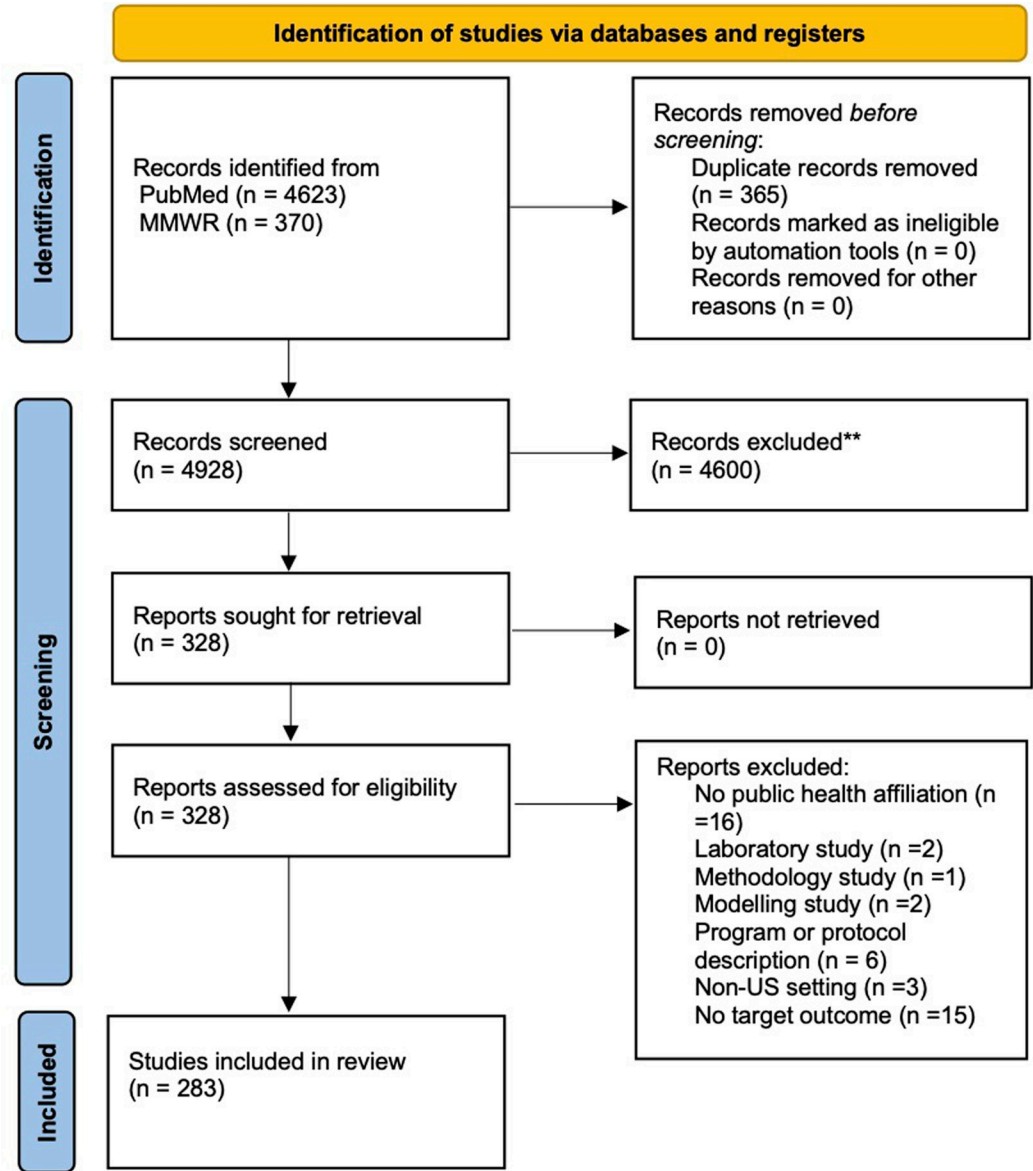

**Fig 1. PRISMA 2020 flow diagram for identification, screening, and inclusion of studies.**

these found that, proportionally, excess mortality in 2020 was highest minority populations and in persons aged 25–44 [269].

Five studies estimated the symptomatic fraction [210, 215, 243, 256, 306]. These studies utilized data from various settings (a military ship, schools, skilled nursing facilities, households, and a prison) and reported heterogeneous results.

Only four analytic studies estimated other infection severity measures. One measured the 30-day probability of fatality for cases in one skilled nursing facility [243], one assessed the infection fatality ratio by ethnicity in New York State [218], and two reported infection hospitalization ratio [261] or hospitalization fatality ratio [296].

Many analytic studies (66/103) evaluated one or more predictor of having a confirmed COVID-19 infection. Specific predictors varied by study and included age or gender [210, 211,

**Table 4. Epidemiologic studies on COVID-19 transmission, infection severity, disease burden with US governmental public health authors affiliations published through November 2021.**

| | Analytic Reports | | | Descriptive Reports | | | |
|---|---|---|---|---|---|---|---|
| | **2020** | **2021** | **Total** | **2020** | **2021** | **Total** | **Total** |
| **All Studies** | 47 | 56 | 103 | 115 | 65 | 180 | 283 |
| **Public Health Affiliation** | | | | | | | |
| Federal | 13 | 19 | 32 | 24 | 16 | 40 | 72 |
| Federal and State or Local | 20 | 22 | 42 | 75 | 35 | 110 | 152 |
| State or Local | 14 | 15 | 29 | 16 | 14 | 30 | 59 |
| **Publication** | | | | | | | |
| Clinical Infectious Diseases | 8 | 2 | 10 | 4 | 2 | 6 | 16 |
| Emerging Infectious Diseases | 1 | 15 | 16 | 8 | 8 | 16 | 32 |
| MMWR | 27 | 17 | 44 | 85 | 45 | 130 | 174 |
| Other Journals | 11 | 22 | 33 | 18 | 10 | 28 | 61 |
| **Data Period** | | | | | | | |
| Jan–Mar 2020 | 7 | 1 | 8 | 28 | 2 | 30 | 38 |
| April–Sept 2020 | 36 | 28 | 64 | 82 | 15 | 97 | 161 |
| Oct 2020–July 2021 | 4 | 24 | 28 | 5 | 42 | 47 | 75 |
| August–Nov 2021 | | 3 | 3 | | 6 | 6 | 9 |
| **Study Design** | | | | | | | |
| Case-control | 2 | 1 | 3 | | | | 3 |
| Case series, cluster, or outbreak | 3 | | 3 | 84 | 41 | 125 | 128 |
| Cross-sectional | 20 | 21 | 41 | | 2 | 2 | 43 |
| Ecologic | 10 | 15 | 25 | | 1 | 1 | 26 |
| Incidence | | | 0 | 31 | 20 | 51 | 51 |
| Prospective | 6 | 2 | 8 | | | | 8 |
| Retrospective | 6 | 17 | 23 | | 1 | 1 | 24 |
| **Principle Source of Study Data** | | | | | | | |
| Active surveillance program | 2 | 3 | 5 | 14 | 3 | 17 | 22 |
| Administrative data records | | 5 | 5 | | | 0 | 5 |
| Original field data | 16 | 6 | 22 | 52 | 25 | 77 | 99 |
| Medical records | 3 | 4 | 7 | 3 | 2 | 5 | 12 |
| Passive surveillance program | 15 | 20 | 35 | 40 | 32 | 72 | 107 |
| Serosurvey | 7 | 13 | 20 | | 1 | 1 | 21 |
| Questionnaire survey | 3 | 3 | 6 | 4 | | 4 | 10 |
| Vital statistics records | 1 | 2 | 3 | 2 | 2 | 4 | 7 |
| **Study Settings** | | | | | | | |
| Assisted living facility | | | 0 | 2 | | 2 | 2 |
| Childcare facility | | | 0 | 2 | | 2 | 2 |
| College or university | 1 | 1 | 2 | 1 | 6 | 7 | 9 |
| Community | 31 | 43 | 74 | 63 | 35 | 98 | 172 |
| Congregate living facilities (multiple) | | | 0 | 1 | 1 | 2 | 2 |
| Correctional or detention facility | 3 | | 3 | 7 | 4 | 11 | 14 |
| Gym or fitness facility | | 1 | 1 | | 2 | 2 | 3 |
| Healthcare facility | 4 | 1 | 5 | 6 | 1 | 7 | 12 |
| Homeless Facility | 1 | 1 | 2 | 3 | | 3 | 5 |
| Military facility | 1 | | 1 | 1 | | 1 | 2 |
| Other Workplace | 2 | 4 | 6 | 4 | 2 | 6 | 12 |
| Primary or secondary school | | 3 | 3 | 1 | 6 | 7 | 10 |

*(Continued)*

**Table 4.** (*Continued*)

| | Analytic Reports | | | Descriptive Reports | | | |
|---|---|---|---|---|---|---|---|
| | 2020 | 2021 | Total | 2020 | 2021 | Total | Total |
| Skilled nursing facility | 4 | 2 | 6 | 16 | 3 | 19 | 25 |
| Social gathering or event | | | 0 | 8 | 5 | 13 | 13 |
| **Study Subpopulations** | | | | | | | |
| Armed Forces | 1 | | 1 | 1 | 1 | 2 | 3 |
| Children and Adolescents | 1 | 9 | 10 | 7 | 12 | 19 | 29 |
| College students | 1 | 1 | 2 | 1 | 5 | 6 | 8 |
| General population | 29 | 37 | 66 | 59 | 37 | 96 | 162 |
| Healthcare Workers or First Responders | 4 | 4 | 8 | 8 | | 8 | 16 |
| Homeless Individuals or Facility Staff | 1 | 1 | 2 | 4 | 1 | 5 | 7 |
| LCTF residents and staff | 4 | 1 | 5 | 20 | 3 | 23 | 28 |
| Other Occupations | 2 | 3 | 5 | 6 | 2 | 8 | 13 |
| Pregnant women | 1 | | 1 | 2 | | 2 | 3 |
| Prison inmates and staff | 3 | | 3 | 7 | 4 | 11 | 14 |
| **Study Outcomes** | | | | | | | |
| Secondary Attack Ratio | 6 | 3 | 9 | | | | 9 |
| Serial Interval / Generation Time | | 2 | 2 | | | | 2 |
| Growth Rate | 1 | 1 | 2 | | | | 2 |
| **Period Incidence Measures** | | | | | | | |
| Any Incidence Measure | 9 | 16 | 25 | 35 | 27 | 62 | 87 |
| Infection Incidence | | 2 | 2 | | | | 2 |
| Case Incidence | 2 | | 2 | 27 | 16 | 43 | 45 |
| ED Visit Incidence | | 1 | 1 | 2 | 1 | 3 | 4 |
| Hospitalization Incidence | | 2 | 2 | 4 | 5 | 9 | 11 |
| Mortality Incidence | | 1 | 1 | 7 | 4 | 11 | 12 |
| Seroprevalence | 8 | 12 | 20 | | 2 | 2 | 22 |
| Excess Deaths | 1 | 2 | 3 | 1 | 2 | 3 | 6 |
| **Predictors of Infection** | | | | | | | |
| Any Infection Predictor | 32 | 34 | 66 | | | | 66 |
| Age or Sex | 15 | 14 | 29 | | | | 29 |
| Race, Ethnicity, or Income | 10 | 11 | 21 | | | | 21 |
| Co-morbidity | 5 | 3 | 8 | | | | 8 |
| Behavioral | 4 | 9 | 13 | | | | 13 |
| Occupational | 7 | 10 | 17 | | | | 17 |
| Environmental | | 2 | 2 | | | | 2 |
| Residential | 8 | 9 | 17 | | | | 17 |
| Prior Infection | | 1 | 1 | | | | 1 |
| Geospatial | 7 | 9 | 16 | | | | 16 |
| **Disease Severity Measures** | | | | | | | |
| Symptomatic Fraction | 4 | 1 | 5 | | | | 5 |
| Case Fatality Ratio | 1 | | 1 | | | | 1 |
| Infection Hospitalization Ratio | | 1 | 1 | | | | 1 |
| Infection Fatality Ratio | 1 | | 1 | | | | 1 |
| Hospital Fatality Ratio | | 1 | 1 | | | | 1 |
| **Predictors of Disease Severity** | | | | | | | |
| Any Severity Predictor | 10 | 13 | 23 | | | | 23 |
| Age or Sex | 4 | 7 | 11 | | | | 11 |

(*Continued*)

**Table 4.** (Continued)

| | Analytic Reports | | | Descriptive Reports | | | |
|---|---|---|---|---|---|---|---|
| | 2020 | 2021 | Total | 2020 | 2021 | Total | Total |
| Race, Ethnicity, or Income | 4 | 6 | 10 | | | | 10 |
| Co-morbidity | 8 | 8 | 16 | | | | 16 |
| Behavioral | | 1 | 1 | | | | 1 |
| SARS-CoV-2 Variant | | 2 | 2 | | | | 2 |
| Geospatial | 1 | 1 | 2 | | | | 2 |

215–217, 219, 222, 224, 226, 235, 237, 239, 242, 244, 253, 254, 257–259, 261, 266, 268, 274, 283–285, 292, 308, 309], race or ethnicity [211, 215, 217, 222, 224, 237, 239, 240, 241, 242, 253, 254, 257, 259, 266, 268, 283, 284, 285, 292, 309], co-morbidities [216, 224, 245, 257, 274, 283, 284, 297], the influence of behaviors such as masking or social distancing [210, 226, 234, 239, 257, 260, 262, 264, 265, 277, 294, 295], occupation, industry or workplace factors [216, 224, 233, 239, 246, 257, 259, 270, 272, 274, 279, 282, 284, 285, 289, 299, 309], and housing status, including homelessness [271, 302] or living in shared spaces like dormitories or detention facilities [236, 300]. Sixteen analytic studies reported on variation of infection incidence by geospatial characteristics including the proportion of particular demographic groups in a community [221, 286, 305, 301, 310], work location [239], neighborhood deprivation/vulnerability levels or income [229, 230, 250, 283, 285], zip code education levels [292], and correlation of school-related infection to community incidence rates [264].

Several studies of infection predictors examined risk factors within specific settings and sub-populations, including face coverings and distancing among occupants of a military ship [210], ethnic composition and exposure risk factors in employees of industrial facilities [222, 241, 265, 270, 282], including one study that compared the incidence before and after mitigation strategies such as masking and barriers in a meat processing facility [265], shelter residence status among people experiencing homelessness [271, 302], screening strategies and staffing levels in skilled nursing facilities [227, 228, 243, 247, 272], housing type and athletic participation among college students [234, 300], and community exposures and symptomatic contacts in children [242, 261, 264, 267, 277, 280, 283, 294, 295, 306]. Several studies use sero-prevalence surveys or administrative data to examine infection risk factors among healthcare workers and first responders [224, 233, 239, 257, 259, 274, 289, 299]. One analytic study evaluated serial testing of healthcare workers in a skilled nursing facility [272].

Twenty-three analytic studies examined predictors for severe disease outcomes, including, most commonly, age, sex, race/ethnicity, and co-morbidities. These studies consistently found older age and co-morbidities to be a strong predictor of need for hospitalization, ICU care, mechanical ventilation and of death. Older age predicted prolonged symptoms among non-hospitalized cases [214]. Two studies looked at the impact of variants on disease severity and hospitalizations [275, 298]. One study compared the risk for in-hospital complications for patients with COVID-19 relative to those with influenza [232].

## Discussion

This scoping review aimed to assess the completeness of authoritative estimates of key epidemiologic data in the United States during the first two years of COVID-19 and the responsiveness of published U.S. governmental public health agency epidemiological research to pandemic knowledge needs. Overall, we found publicly available authoritative estimates for most expected transmission and disease severity parameters; however, some were lacking, and

others had significant uncertainties. While official US CDC estimates of these parameters appeared consistent across time-periods, we observed limited assessment of these parameters in US populations as well as a lack of re-assessment over the course of the pandemic.

Nationally standardized measures of infection and disease incidence published by the US CDC had limited resolution through most of 2020; however, by the end of 2021, the US CDC was disaggregating most surveillance indicators by county geography and sex, age, and race/ethnicity. Resolution with regards to sub-populations (e.g., occupational groups, those with prior infection) and specific exposure settings (e.g., workplaces, congregate facilities) remained lacking.

Investigators affiliated with US government public health entities published a large volume of epidemiological reports. The majority, however, were either descriptive studies such as cluster investigations or reports of period and population-specific case incidence. Descriptive studies, while useful for hypothesis generation, will not provide sufficiently reliable or generalizable information for designing or evaluating mitigation strategies.

Collectively, analytic studies published in this period were numerous but did not address key knowledge gaps. We discuss these knowledge gaps and their significance further below.

## Gaps in infectious transmission and disease severity data

Transmission parameters, including the incubation period, serial interval, the clinical fraction, and timing and duration of infectiousness are valuable for predicting the pace and magnitude of epidemic growth and for establishing the feasibility and efficacy of isolation, quarantine and contact tracing practices.

Estimates of the symptomatic (clinical) fraction for COVID-19 had a wide confidence range, likely reflecting significant heterogeneity among study methods, settings, and populations [311–314]. Ongoing estimation of the clinical fraction utilizing long term cohort studies might have informed understanding of evolving population immunity as well as the pathogenicity of COVID-19 variants.

We found no U.S. authoritative estimates of the secondary attack rate (SAR) either for households or other community settings. International meta-analytic reviews of the SAR did provide summary estimates of the SAR for household settings as well as for subpopulations within households [315]. These reviews did include several US based studies. Additional US studies estimating the SAR for household, community, business, transportation, and educational settings might have informed public understanding of comparative setting-specific transmission risks and might have focused attention on additional policy interventions, such as the provision of medical isolation housing.

Authoritative estimates of population susceptibility to COVID-19–100% at the pandemic's onset–did not change over the first 18 months of the pandemic. While the US CDC estimated a cumulative 146 million COVID-19 infections occurred in the US as of September 2021 [26], the duration and clinical significance of infection-provided immunity remained poorly characterized [316]. In addition, national publicly available surveillance data was not disaggregated by prior infection status.

Many studies have assessed vulnerability factors for severe disease, hospitalization. However, authoritative estimates of the infection fatality ratio (IFR) based on European countries may not have been generalizable to the US given country-level differences in infection ascertainment, co-morbidities, social vulnerability, and medical care. IFR estimates were also not re-evaluated over the course of the pandemic despite the rapid evolution of clinical therapeutics. Establishing large scale community cohort studies in multiple regions might have supported ongoing assessment of IFR as well as other infection severity parameters.

## Gaps in infection and disease burden surveillance data

Further disaggregation of surveillance measures may have optimized the targeting and timing of community mitigations. Through the National Notifiable Disease Surveillance System (NNDSS), CDC accumulated tens of millions of surveillance case reports. Case report forms included data fields for exposure information on residence, occupation, travel [317]. However, except for skilled nursing facilities and healthcare personnel, national surveillance data was not reported by exposure setting, exposure history, or occupation. Complete and consistent collection and reporting of data elements in standard surveillance case reports might have improved understanding of the relative burden of infection across settings and modes of contact.

Geographically specific data on COVID-19 hospital admissions also lagged. Through most of 2020, CDC published estimates of age-stratified hospital admissions incidence only from active surveillance in 13 sub-state regions. The U.S. Centers for Medicare Services (CMS) first issued requirements for COVID-19 hospital admission reporting in July 2020, and standardized regional data on hospitalization admissions first became available in December 2020.

Notably, case counts of laboratory confirmed infection remained the dominant indicator of pandemic dynamics despite well-understood recognition that case counts underestimate infection incidence variably across population, place, and time [5, 318]. The pre-COVID US National Pandemic Strategy envisioned transitioning from counting individual confirmed cases to monitor epidemic trends to monitoring illness rates (i.e., hospitalization admissions and syndromic surveillance) [2]. During the H1N1 pandemic, the USCDC discontinued state reporting of individual lab-confirmed cases two months after the initiation of the epidemic and initiated state reporting of total numbers of weekly H1N1 hospitalizations and deaths [4, 319].

## Alignment between COVID-19 science and policy questions

Governments implemented novel and controversial policies to mitigate the COVID-19 pandemic, such as stay at home orders, school and business closures, and mask mandates. While "precautionary" these polices had little *a priori* high-quality supporting evidence [320]. Implementing novel policies raise difficult questions of societal trade-offs and demand timely real-world evaluation (Table 5). However, much of the research reported by US public health entities has had little direct bearing on specific pandemic mitigation policy and practices.

Research previously conducted during the H1N1 pandemic in the US as well as examples of COVID-19 research conducted by non-governmental actors and peer countries suggest that US public health research efforts might have done more to evaluate community mitigation policies. During the 2009 influenza H1N1 pandemic, for example, multiple transmission studies conducted by US CDC investigators contributed to timely estimates of the SAR and their determinants, including age, setting, and timing [321, 322].

**Table 5. Examples of research questions relevant to US COVID-19 policy debates.**

| |
|---|
| What is the optimum duration of isolation and quarantine? |
| Are isolation and quarantine effective mitigation strategies, given asymptomatic and pre-symptomatic transmission? |
| What is the relative share of disease attributable to different community settings (households, workplaces, retail, transport, health care, schools) |
| How effective are masks and face coverings for preventing transmission in different settings |
| Are safety measures in essential workplaces and public transport adequate to prevent occupational transmission? |
| What are health costs and benefits of closing schools? |
| How well does recovery from infection protect against subsequent infection and severe disease? |

Systematic reviews provide another lens on the scope of US COVID-19 research contributions. A systematic review of mask effectiveness published in late 2020 included only one small U.S. study in health workers [323]. A meta-analysis of 61 studies on COVID-19 workplace prevention measures included 15 US studies which were limited to healthcare and skilled nursing settings [324]. None of the 11 studies included in a December 2020 meta-analysis on transmission of COVID-19 by children in schools were set in the US [325]. A November 2021 review identified several large U.S. studies on infection-derived protective immunity; none had U.S. government affiliations [326].

Other countries appeared to better leverage public surveillance to systematically assess vulnerable population subgroups. Norwegian public health authorities estimated comparative infection risks by occupation across different phases of the pandemic identifying health care, food service, transportation, childcare and teaching as risky settings [327]. UK scientists similarly used published national statistics to estimate age-standardized COVID-19 mortality incidence among occupations finding significantly higher mortality among taxi drivers, low skilled occupations, and personal care workers [328].

## Limitations

This scoping review has several limitations. We recognize that clinical, academic, and other private institutions in the U.S. also made substantial contributions to COVID-19 data and research. Nevertheless, we limited our review to governmental public health data and research for the following reasons: (1) State and federal public health agencies are the responsible entities for infectious disease surveillance, including collecting, compiling, cleaning, standardizing, and interpreting data; (2) Only public agencies receive legislatively mandated communicable disease reports and have the authority to conduct disease investigations; (3) Authoritative public health data and guidance underpins public policy decisions.

Our search strategy may have missed relevant published reports and we did not consider pre-prints or unpublished agency analyses. Furthermore, we did not examine or judge data or study quality, including precision and bias.

## Explanations for knowledge gaps

Understanding the causes of these observed knowledge gaps could support planning for future pandemics. In the case of COVID-19, reasons may have been due both to institutional capacities (e.g., time, resources, data, methodological feasibility) as well as to organizational priorities and leadership choices.

Limited availability of timely comprehensive and standardized data may have been a contributing factor [329, 330]. In the US, local and state agencies have primary and statutory responsibility for collecting and organizing infectious disease surveillance data. State statutes typically require health care providers or laboratories to report incident cases; public health investigators subsequently conduct case interviews to elaborate on the context of exposure. However, local and state public health agencies implement disease control activities using heterogenous practices and with varying capacities [331]. The US CDC reported significant variation in the timeliness and quality of federal reporting by state authorities [332]. State by state online public reporting of COVID-19 data was also deficient [333]. Further, federal health oversight agencies were slow to require standardized disease data from hospitals and skilled nursing facilities.

Failures to identify and communicate with contacts during disease control investigations also limited the utility of data that might have come from contact-tracing. Surveys of health departments during the pandemic reported that case investigators conducted timely COVID-

19 case interviews on only a fraction of incident case, identifying and reaching fewer than one contact per case [334, 335].

While many epidemiologic questions about COVID-19 required well-established research methods, the rapid and simultaneous implementation of numerous non-pharmaceutical interventions created methodological challenges for researchers. In the case of school closure for example, an international systematic review concluded a lack of evidence of a significant protective effect, finding that reviewed studies frequently suffered from unaddressed biases from confounding and collinearity from other non-pharmacological interventions [336].

Institutional priorities and choices on the research agenda might also have had influence. It remains unclear whether research conducted by US public health investigators was systematically coordinated or directed. The CDC first publicly offered a set of COVID-19 research priorities in March 2021—one year after the start of the pandemic [337].

Greater transparency of data may have facilitated more timely corrective responses. The CDC has publicly acknowledged selectively releasing the data it has collected [338]. Some states also resisted calls for full public reporting of covid surveillance data [339].

## Conclusions

In conclusion, over the first eighteen months of the COVID-19 pandemic, public health authorities in the U.S. appear to have lacked complete and timely epidemiological data for optimal disease control responses. These gaps occurred despite pre-established pandemic data collection priorities and significant public resource commitments to COVID-19 pandemic response. We observed, for example, a delayed implementation of standardized national COVID-19 surveillance measures and limited validation and re-assessment of essential transmission and infection severity parameters. US public health scientists authored many original publications on COVID-19; however, most were descriptive, and few provided high-quality evidence to inform salient policy and management decisions.

Moving forward, U.S. public health agencies should examine the reasons for these gaps and plan for a timely, strategic, and prioritized national epidemiological data collection and research agenda for future rapidly emerging infectious disease epidemics. Improved data-driven responses may require national standards for disease control data collection and management, ready-to-use research protocols, and a greater commitment to publicly transparency.

## Supporting information

**S1 Protocol. Epidemiological data for COVID-19 pandemic management in the United States: A protocol for a scoping review.**
(PDF)

**S1 Table. Preferred Reporting Items for Systematic reviews and Meta-Analyses extension for Scoping Reviews (PRISMA-ScR) checklist.**
(PDF)

**S2 Table. Abstracted data from reports meeting scoping review inclusion criteria.**
(PDF)

## Author Contributions

**Conceptualization:** Rajiv Bhatia.

**Data curation:** Rajiv Bhatia, Isabella Sledge.

**Formal analysis:** Rajiv Bhatia, Isabella Sledge.

**Investigation:** Rajiv Bhatia, Isabella Sledge.

**Methodology:** Rajiv Bhatia, Isabella Sledge.

**Writing – original draft:** Rajiv Bhatia, Isabella Sledge.

**Writing – review & editing:** Stefan Baral.

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
