## [Decision Letter · Decision Letter 0]

29 Oct 2021

PONE-D-21-07710The Missing Science: Epidemiological data gaps for COVID-19 policy in the United StatesPLOS ONE

Dear Dr. Bhatia,

Thank you for submitting your manuscript to PLOS ONE. After careful consideration, we feel that it has merit but does not fully meet PLOS ONE’s publication criteria as it currently stands. Therefore, we invite you to submit a revised version of the manuscript that addresses the points raised during the review process.

The reviewer recommends that you make substantial revisions to the manuscript. Please attend to all the concerns raised and return in the revised manuscript as advised in this letter.

We look forward to receiving your revised manuscript.

Kind regards,

Martin Chtolongo Simuunza, PhD

Academic Editor

PLOS ONE

Journal Requirements:

2. Please upload with your submission a completed copy of the PRISMA checklist (available from: https://prisma-statement.org/PRISMAStatement/Checklist) as Supplementary Information.

3. . Please include captions for your Supporting Information files at the end of your manuscript, and update any in-text citations to match accordingly. Please see our Supporting Information guidelines for more information: http://journals.plos.org/plosone/s/supporting-information.

Reviewers' comments:

Reviewer's Responses to Questions

**Comments to the Author**

1. Is the manuscript technically sound, and do the data support the conclusions?

Reviewer #1: Yes

2. Has the statistical analysis been performed appropriately and rigorously? 

Reviewer #1: N/A

3. Have the authors made all data underlying the findings in their manuscript fully available?

Reviewer #1: Yes

4. Is the manuscript presented in an intelligible fashion and written in standard English?

Reviewer #1: Yes

5. Review Comments to the Author

Reviewer #1: This is a much needed and important article. The public health data collection efforts in

the United States have been incredibly fragmented. This, though, leads me to my main

critiques of the paper.

Overall

The authors state in the conclusion that:

“CDC scientists have the access to data, the expertise, and the resources to provide the

data necessary for an optimal epidemic response.”

This is actually not entirely true. Public health data collection in the United States is

heavily fragmented. In fact, many states did not report or provide the CDC with their

COVID-19 surveillance data. This was further exacerbated by the lack of data sharing

between health systems and hospitals, as well as, between hospitals and departments

of public health. I think that adding a more nuanced, in-depth discussion of why the

CDC might not have updated domestic data (or why we still have COVID-19 data

problems over a year later) could make this article stronger. Despite the July 2020 HHS

requirement, data collection was in fact a bit of a mess and extremely cumbersome.

Refer to (pg. 13 provides a few examples): https://oig.hhs.gov/oei/reports/OEI-09-21-

00140.pdf

Additionally see: Galaitsi, S.E., Cegan, J.C., Volk, K., Joyner, M., Trump, B.D. and

Linkov, I., 2021. The challenges of data usage for the United States’ COVID-19

response. International Journal of Information Management, 59, p.102352.

Perhaps, the CDC should place more thought into how data is collected and who is in

charge of establishing the back end for data collection. Alternatively, how do you deal

with situations where state and federal requirements do not align but instead cause

increased administrative burdens?

If fact, it might explain a very well received point the authors make: “The content of their

[the CDC’s] investigations raises questions about whether and how an explicit national

research agenda guided CDC epidemiological endeavors…” as well as, “Regional

differences in prevalence could explain part of the variation in the pace of infectious

spread.” Politics and federalism clearly play/played a role in data collection and

reporting (Florida and Nebraska anyone?), as has been discussed in multiple published

articles this last year. For example, see:

Rocco, P., Rich, J.A., Klasa, K., Dubin, K.A. and Béland, D., 2021. Who Counts Where?

COVID-19 Surveillance in Federal Countries. Journal of Health Politics, Policy and Law.

Additionally, the authors could suggest (and cite existing literature) about how to best go

about data collection for the specific parameters that they identify. For example, are

there essential occupations that we know have had more exposure risk or that peerreviewed

articles have shown to have higher transmission rates? Is there perhaps a

region, state, or city that was a stellar example of data collection and data-informed

policy decisions? What did they do correct?

I think that more thought should be placed on who is collecting the data (or should be in

charge of collecting the data). The authors infer that this is the CDC, but what

challenges might the CDC face in collecting data? And if it is inevitably going to face

difficulties then how can we make data collection most effective (and ensure health

equity)? What does the evidence say that worked this last year and what didn’t work?

Right now, this reads like a giant list of wants in a perfect world. But, these data might

not all be possible (or feasible or cost effective) to collect.

Discussion (general)

I think that the authors could cite existing studies in peer-reviewed literature that do

exist whenever they mention that “x” doesn’t exist or “y” doesn’t exist. I realize that the

authors limited their search to only studies with CDC affiliated authors, but I think that it

is important to emphasize that local (city, county) and state efforts did exist that tried to

figure this data out because the CDC was very slow to provide good data throughout

2020. There were studies conducted on infectious transmission, disease severity,

burden of infection/disease in smaller settings since COVID-19 began (just not with

CDC affiliated investigators). It might be important to cite a few of these throughout the

discussion and note that the CDC might have determined that the few US studies that

do exist were too small for them to consider. There were some universities that

partnered with state or local departments of public health that did some good work.

Infectious Transmission Section

• Could you cite a few studies that do exist that tried to quantify infectious

transmission? It would be useful to reference what the ‘best available” study that

exists is (for example, ‘no identified transmission studies have quantitatively

examined….then what are the qualitative studies that exist?)

• Are the restaurant restrictions not because of the believe that the virus is

airborne?

• Would state-to-state variability in policy matter at the community level as much

as county-to-county variation?

• Many employers do not report COVID-19 infections.

• Contact tracing will require community trust – even in places with robust contact

tracing efforts, follow-up was often difficult or challenging (especially in lowincome

or immigrant communities)

• Some household transmission studies were conducted. For example, Lewis,

N.M., Chu, V.T., Ye, D., Conners, E.E., Gharpure, R., Laws, R.L., Reses, H.E.,

Freeman, B.D., Fajans, M., Rabold, E.M. and Dawson, P., 2020. Household

transmission of SARS-CoV-2 in the United States. Clinical infectious diseases:

an official publication of the Infectious Diseases Society of America.

Disease Severity Section

• In the second sentence, did you mean “county to country” differences or “county

to county” differences or “country to country” differences?

Minor Edits

Last, there are some minor edits that need to be made:

• There are a few grammatical errors throughout that shouldn’t be too difficult to fix

after another read through

• Citations need to be fixed and standardized

• Make sure to add updated citations and engage with more recent literature

6. PLOS authors have the option to publish the peer review history of their article (what does this mean?). If published, this will include your full peer review and any attached files.

Reviewer #1: No

---

## [Author Response · Author response to Decision Letter 0]

8 Jul 2022

Authors’ Response to Reviewer 1

C1: The authors state in the conclusion that: “CDC scientists have the access to data, the expertise, and the resources to provide the data necessary for an optimal epidemic response.”This is actually not entirely true. Public health data collection in the United States is heavily fragmented. In fact, many states did not report or provide the CDC with their COVID-19 surveillance data. This was further exacerbated by the lack of data sharing between health systems and hospitals, as well as, between hospitals and departments of public health. I think that adding a more nuanced, in-depth discussion of why the CDC might not have updated domestic data (or why we still have COVID-19 data problems over a year later) could make this article stronger. Despite the July 2020 HHS requirement, data collection was in fact a bit of a mess and extremely cumbersome. Refer to (pg. 13 provides a few examples): https://oig.hhs.gov/oei/reports/OEI-09-21-

00140.pdf

Additionally see: Galaitsi, S.E., Cegan, J.C., Volk, K., Joyner, M., Trump, B.D. and

Linkov, I., 2021. The challenges of data usage for the United States’ COVID-19

response. International Journal of Information Management, 59, p.102352.

Perhaps, the CDC should place more thought into how data is collected and who is in

charge of establishing the back end for data collection. Alternatively, how do you deal

with situations where state and federal requirements do not align but instead cause

increased administrative burdens?

If fact, it might explain a very well received point the authors make: “The content of their

[the CDC’s] investigations raises questions about whether and how an explicit national

research agenda guided CDC epidemiological endeavors…” as well as, “Regional

differences in prevalence could explain part of the variation in the pace of infectious

spread.” Politics and federalism clearly play/played a role in data collection and

reporting (Florida and Nebraska anyone?), as has been discussed in multiple published

articles this last year. For example, see:

Rocco, P., Rich, J.A., Klasa, K., Dubin, K.A. and Béland, D., 2021. Who Counts Where?

COVID-19 Surveillance in Federal Countries. Journal of Health Politics, Policy and Law.

Additionally, the authors could suggest (and cite existing literature) about how to best go

about data collection for the specific parameters that they identify. For example, are

there essential occupations that we know have had more exposure risk or that peer reviewed

articles have shown to have higher transmission rates? Is there perhaps a

region, state, or city that was a stellar example of data collection and data-informed

policy decisions? What did they do correct?

I think that more thought should be placed on who is collecting the data (or should be in

charge of collecting the data). The authors infer that this is the CDC, but what

challenges might the CDC face in collecting data? And if it is inevitably going to face

difficulties then how can we make data collection most effective (and ensure health

equity)? What does the evidence say that worked this last year and what didn’t work?

Right now, this reads like a giant list of wants in a perfect world. But, these data might

not all be possible (or feasible or cost effective) to collect.

R1: We have added a new sub-section in the discussion to discuss possible explanations for the gaps we observed in public health agency surveillance data and original investigations dependent on that data which includes discussion of the lack of standardized surveillance data collection and reporting protocols. We note, however, that many epidemiological investigations utilized data sources that not derived from public disease surveillance systems. And many of the research gaps we identify might have been addressed with the application of well-established method (e.g. community cohort studies). 

C2: I think that the authors could cite existing studies in peer-reviewed literature that do

exist whenever they mention that “x” doesn’t exist or “y” doesn’t exist. I realize that the

authors limited their search to only studies with CDC affiliated authors, but I think that it

is important to emphasize that local (city, county) and state efforts did exist that tried to

figure this data out because the CDC was very slow to provide good data throughout

2020. There were studies conducted on infectious transmission, disease severity,

burden of infection/disease in smaller settings since COVID-19 began (just not with

CDC affiliated investigators). It might be important to cite a few of these throughout the

discussion and note that the CDC might have determined that the few US studies that

do exist were too small for them to consider. There were some universities that

partnered with state or local departments of public health that did some good work.

R2: We concur. We expanded the scope of our review to include all published reports affiliated either with the US CDC or with any US state or local public health agencies. Most included studies were conducted and authored by collaborations of federal, state, and local governmental public health actors. We also added a new sub-section in the discussion that describes the contributions of U.S. public health affiliated publications to issue-specific COVID-19 epidemiological reviews. 

C3: Could you cite a few studies that do exist that tried to quantify infectious

transmission? It would be useful to reference what the ‘best available” study that

exists is (for example, ‘no identified transmission studies have quantitatively

examined….then what are the qualitative studies that exist?)

R3. We included several U.S. public health affiliated studies that quantified the household secondary attack rate in the revision. We referenced a review of studies of the household secondary attack rate that demonstrates the international scope of research on the topic.

C4: Are the restaurant restrictions not because of the believe that the virus is

airborne?

R4: We removed the mention on restaurant restrictions specifically from the revised manuscript. We note that restrictions on in-person restaurant dining occurred before there existed either public health agency acknowledgement of airborne transmission. 

C5: Would state-to-state variability in policy matter at the community level as much

as county-to-county variation?

R5: We agree that studying the variation local policy would be optimal. 

C6: Many employers do not report COVID-19 infections.

R6: We agree. However, health care facilities and laboratory providers are subject to statutory requirements report cases under U.S. law. Employers do not have this statutory responsibility nor the ability, in most cases, the ability to assess COVID-19 infection status among their employees. However, case surveillance reports submitted by healthcare and utilized by public health agencies do include fields for occupation, industry, and work location. We acknowledge that morbidity report forms submitted to State and Federal public health agencies varied in completeness.

C7: Contact tracing will require community trust – even in places with robust contact

tracing efforts, follow-up was often difficult or challenging (especially in low income

or immigrant communities)

R7: We agree. In the revision, we have noted the challenges public health agencies faced in gaining participation from cases and contacts during disease investigations.

C8: Some household transmission studies were conducted. For example, Lewis, N.M., Chu, V.T., Ye, D., Conners, E.E., Gharpure, R., Laws, R.L., Reses, H.E., Freeman, B.D., Fajans, M., Rabold, E.M. and Dawson, P., 2020. Household transmission of SARS-CoV-2 in the United States. Clinical infectious diseases: an official publication of the Infectious Diseases Society of America.

R8: The above study by Lewis et al. was included in our initial review. We include several US additional household transmission studies in the updated review. 

C9: In the second sentence, did you mean “county to country” differences or “county

to county” differences or “country to country” differences?

R9: We have revised the sentence to clarify. 

C10: There are a few grammatical errors throughout that shouldn’t be too difficult to fix

after another read through

R10: Noted.

C11: References are in non-standard formats.

R11: All References are now standardized to NLM formats

C12: Make sure to add updated citations and engage with more recent literature

R12: The revision extends the review of authoritative estimates and published public health affiliated reports through Nov 30, 2021.

---

## [Decision Letter · Decision Letter 1]

12 Sep 2022

Missing Science: A Scoping Study Of COVID-19 Epidemiological Data in the United States

PONE-D-21-07710R1

Dear Dr. Bhatia,

We’re pleased to inform you that your manuscript has been judged scientifically suitable for publication and will be formally accepted for publication once it meets all outstanding technical requirements.

Kind regards,

Martin Chtolongo Simuunza, PhD

Academic Editor

PLOS ONE

Additional Editor Comments (optional):

Reviewers' comments:

Reviewer's Responses to Questions

**Comments to the Author**

1. If the authors have adequately addressed your comments raised in a previous round of review and you feel that this manuscript is now acceptable for publication, you may indicate that here to bypass the “Comments to the Author” section, enter your conflict of interest statement in the “Confidential to Editor” section, and submit your "Accept" recommendation.

Reviewer #1: All comments have been addressed

2. Is the manuscript technically sound, and do the data support the conclusions?

Reviewer #1: Yes

3. Has the statistical analysis been performed appropriately and rigorously? 

Reviewer #1: N/A

4. Have the authors made all data underlying the findings in their manuscript fully available?

Reviewer #1: Yes

5. Is the manuscript presented in an intelligible fashion and written in standard English?

Reviewer #1: Yes

6. Review Comments to the Author

Reviewer #1: Thank you so much for letting me review this paper. I can tell that a huge amount of effort and work went into your revisions. Table 3 and Table 4 are excellent! Table 5 could be a table or a figure (if a figure is easier to make with just the list of questions in a box or something like that). The paper is excellent, informative, and a great contribution to public health research! I do not think that any more substantive edits need to be made. Fantastic job :)

If there is time, then there are only a few minor grammar edits to be made throughout. A quick read through with fresh eyes should catch all of them.

For example:

Page 6, paragraph 2: “….Our aims were to assess the completeness of this data during the first two years of the pandemic-19…”

I believe you wanted to stay “COVID-19 pandemic” or “pandemic caused by the Sars-COV-2 virus”.

Page 25, paragraph 3: “Furthermore, also did not examine or judge…”

I believe you wanted to say “we also did not examine…”

Page 26, paragraph 3: “While many epidemiologic questions COVID-19 required….”

I believe you wanted to say “While many epidemiologic questions about COVID-19…”

7. PLOS authors have the option to publish the peer review history of their article (what does this mean?). If published, this will include your full peer review and any attached files.

Reviewer #1: No

---

## [Editor Report · Acceptance letter]

29 Sep 2022

PONE-D-21-07710R1 

Missing Science: A Scoping Study Of COVID-19 Epidemiological Data in the United States 

Dear Dr. Bhatia:

I'm pleased to inform you that your manuscript has been deemed suitable for publication in PLOS ONE. Congratulations! Your manuscript is now with our production department. 

Kind regards, 

on behalf of

Dr. Martin Chtolongo Simuunza 

Academic Editor

PLOS ONE